# An In Vitro Approach to Prime or Boost Human Antigen-Specific CD8^+^ T Cell Responses: Applications to Vaccine Studies

**DOI:** 10.3390/vaccines13070729

**Published:** 2025-07-04

**Authors:** Hoang Oanh Nguyen, Mariela P. Cabral-Piccin, Victor Appay, Laura Papagno

**Affiliations:** 1Univ. Bordeaux, CNRS, Inserm, ImmunoConcEpT, UMR 5164, ERL 1303, F-33000 Bordeaux, France; hoang-oanh.nguyen@u-bordeaux.fr (H.O.N.); victor.appay@u-bordeaux.fr (V.A.); 2International Research Center, A.C.Camargo Cancer Center, São Paulo 01508-010, Brazil; mariela.piccin@accamargo.org.br

**Keywords:** vaccination, immunomonitoring, T cells, priming, flow cytometry

## Abstract

Although vaccine development has primarily focused on inducing neutralizing antibodies, increasing evidence supports an important role of CD8^+^ T cell responses in vaccine effectiveness. Routine assays, which are mainly based on antibody titers, may therefore not accurately reflect the full immune response elicited by vaccination. Assessing antigen-specific T cell responses upon vaccination poses several challenges. A common issue in studying T cells specific to a vaccine antigen is their low frequency in circulation, which can limit their ex vivo analysis. Moreover, the use of human cell-based models is crucial for studying and optimizing the induction of T cell responses to design effective vaccines. We developed an innovative in vitro approach of human CD8^+^ T cell priming, based on the rapid mobilization of dendritic cells (DCs) directly from unfractionated peripheral blood mononuclear cells (PBMCs). This simple and original method allows for side-by-side comparisons of multiple test parameters in a standardized system, providing both quantitative and qualitative readouts of primed antigen-specific CD8^+^ T cells. Here, we discuss the genesis of this approach and its versatile applications, including monitoring antigen-specific T cell responses, evaluating an individual’s T cell priming capacity, and conducting preclinical studies on potential adjuvants and vaccine candidates.

## 1. Introduction

### 1.1. Need for Human-Based Approaches and Assays in Vaccinology

Vaccines are among the most useful means of preventive medicine. Their administration can protect a healthy person from getting infected by a harmful pathogen, but can also help to reduce hospitalization or mortality rate in the case of infection [1]. In theory, both humoral immunity and cellular immunity are important to protect people from pathogens or diseases. In practice though, the majority of current licensed vaccines mainly induce antibodies against specific pathogens, along with helper CD4^+^ T cells, but few CD8^+^ T cells. CD8^+^ T cells are nonetheless essential for efficacious viral clearance [2,3,4,5,6,7], as they induce apoptotic cell death of infected cells upon recognizing pathogen derived peptides presented by MHC class I molecules on the target cell surface via their T cell receptors (TCRs) [8]. They are also central players of anti-tumoral immune responses, and inducing these cells represents a primary goal of cancer immunotherapies, including vaccines. CD8^+^ T cell activities are characterized by the release of several effector cytokines (e.g., IFN-γ, TNF, IL-2) to support host defense and various powerful cytotoxic mechanisms (exocytosis of lytic granules, Fas ligand) to kill target cells [9]. Inducing both humoral- and cell-mediated immunity arms is certainly important for vaccine efficacy and evaluating these two arms provides important insights into protection.

Vaccine development and validation involve multiple phases including testing a vaccine platform in pre-clinical studies and monitoring its immunogenicity in clinical studies. These two steps are critical to designing efficacious vaccines [10] but present important complications. The mouse model is the main experimental in vivo model to test immune responses to candidate vaccines. However, biological differences between mice and men render it difficult to perform comparative extrapolations and reach conclusive predictions as to what may happen in humans [11]. Certain pathways, receptors, or cytokines are non-existant or non-functional in mice compared to humans. For instance, signaling through TLR8, which recognizes single-stranded-RNA and induces strong inflammatory signals in human cells, does not function equivalently in mice [12]. Assessing the immunostimulatory effects of this pathway for vaccine development faces challenges due to the lack of appropriate in vivo models. Moreover, approaches that could take into consideration the biological diversity of humans and pathological settings for the study of T cell responsiveness would be extremely useful. For instance, the quantity and quality of T cells or antigen presenting cells (APCs) can be strongly altered in older people or immunodeficient individuals, weakening immune responses upon antigenic stimulation [13]. A pre-clinical approach using human material may be useful in these different contexts.

Regarding the analysis of vaccine immunogenicity, the magnitude and the effectiveness of humoral responses are often measured following immunization or infections and mainly determined by the level of neutralizing antibodies generated by B cells. Taking into account the important role of CD8^+^ T cells in targeting cell elimination, and the growing interest in T cell-based vaccines and therapies, monitoring CD8^+^ T cell responses is essential for vaccine screening and development [2,9]. There are a number of methods to measure T cell immunity ex vivo, including Elispot, intracellular cytokine staining (ICS), or peptide–MHC multimer complexes [11,14,15]. Elispot and ICS rely on the production of cytokines by specific T cells [16,17] but the frequency of cytokine-producing cells may be rare so that these approaches may underestimate the frequency of antigen-specific T cells [15,18]. Peptide–MHC multimers enable direct detection of antigen-specific T cells but are limited to certain peptide–MHC complexes. Standard approaches can be problematic when frequencies of T cells specific for a given antigen are low [10,15], as this can undermine the accuracy of results. A strategy to enable the detection of rare antigen-specific T cells is the use of in vitro approaches that include additional steps of cell amplification.

A human-based assay capable of studying the antigen-specific T cell priming process and amplifying T cell frequencies could help overcome some of these limitations. In this review, we discuss the development of such an in vitro approach and explore its various applications in vaccine research, including monitoring antigen-specific T cell responses, assessing individuals’ T cell priming capacities, and screening for potential adjuvants and vaccine candidates.

### 1.2. Genesis of a Multipurpose CD8^+^ T Cell Induction Approach

In vitro approaches to expand human T cells are not novel but can vary in efficacy and practicality. Promoting antigen presentation by dendritic cells, along with their stimulatory properties, is a valuable strategy for enhancing T cell responses, thereby facilitating their detection and analysis. Traditional protocols for dendritic cell induction involve the purification and differentiation of blood monocytes, followed by a maturation step using proinflammatory stimuli [19]. Greenberg and colleagues pioneered a protocol called Antigen-Specific Activation and Priming of human T cells, utilizing peptide-loaded monocyte-derived dendritic cells (moDCs) and purified naive CD8^+^ T cells [20]. This approach is suitable for investigating the immunogenicity of peptides and the impact of various molecules on the priming process following 10 days of co-culture with APCs and T cells. However, this approach is labor-intensive due to the complicated co-culture setting, related to the steps of monocyte and naive T cell enrichment and of moDC preparation. Moreover, it is limited to antigen presentation by moDCs, which can differ from the capacity and complexity of having other APCs. Lastly, this assay requires a large initial number of PBMCs to perform the cell purification. In parallel, Mallone and colleagues developed a protocol called accelerated cocultured DC (acDC) assay, designed to detect low frequencies of antigen-specific memory T cells [21]. This approach utilizes specific factors to stimulate dendritic cells directly from whole blood or total PBMCs, together with proteins or peptides for 48 h to activate antigen-specific T cells. It offers several advantages, notably the use of total PBMCs, which contain most of the immune components necessary to induce T cell responses. Moreover, the protocol is straightforward and requires only a relatively limited quantity of PBMCs, which is a critical consideration for longitudinal monitoring of T cell responses, especially in pediatric studies, and a significant benefit when screening peptide libraries.

By combining the advantages of the two approaches (i.e., stimulation of dendritic cells in total PBMCs and priming of antigen-specific CD8^+^ T cells), we could innovate by designing a simple and versatile in vitro system to effectively induce and monitor human antigen-specific CD8^+^ T cell responses from the naive T lymphocyte compartment. To ensure that sufficient numbers of antigen-specific CD8^+^ T cells were present in the naive pool for priming, we used the Melan-A/MART-1 epitope (EAAGIGILTV26-35) as a model antigen, since it is recognized at remarkably high precursor frequencies in HLA-A*0201^+^ (HLA-A2^+^) individuals [22,23]. This large specific TCR repertoire in the naive CD8^+^ T cell compartment provides a useful tool for the study of human antigen-specific T cell priming. To ensure optimal immunogenicity, we used the heteroclitic sequence of the epitope: ELAGIGILTV (ELA) [24]. Moreover, the ELA epitope was incorporated into a 20 mer synthetic long peptide (SLP, ELA-20) as a means of limiting antigen display to DCs with cross-presentation capacity [25,26,27,28]. Focusing on a specific epitope enables the detection of the primed cells with peptide–MHC multimers, and therefore to perform fine functional characterizations of these cells (e.g., expression of cytotoxins or secretion of effector cytokines). To study recall CD8^+^ T cell responses, antigens ranging from optimized CTL epitopes (e.g., the HLA-A2 restricted influenza virus matrix GIL or SARS-CoV-2 spike YQL epitopes, or various HLA-I restricted HIV-1 epitopes) to overlapping peptides (e.g., covering the SARS-CoV-2 spike protein) can also be used to stimulate PBMCs from influenza, SARS-CoV-2- or HIV-1-infected subjects. Overall, this system enables us to study simultaneously the frequency as well as functional parameters of antigen-specific CD8^+^ T cells expanded from the naive or memory T cell compartments of unfractionated human PBMCs in response to cognate antigens [29]. The protocol of this approach and its applications are described below (see Figure 1).

### 1.3. Description of the Standard Protocol

Reagents:Media: AIM-V serum-free medium and R10 medium (RPMI-1640 supplemented with antibiotics (1%), L-glutamine (1%), and FCS (10%)).Cells: Fresh or frozen PBMCs from HLA-A2^+^ donors.DC modulation factor: FLT3L: stock at 10 μg/mL. Working concentration is 50 ng/mL.Antigens: Peptides for de novo (e.g., ELA Melan-A peptide) or recall (e.g., GIL influenza peptide) responses, reconstituted at 10 mg/mL in DMSO, to be used at a final concentration of 1 μg/mL or 0.01 μg/mL, respectively.Stimulating agents: PRR ligands, cytokines…

Cell preparation and stimulation:•Day 0: Mobilizing dendritic cells with FLT3L -Resuspend PBMCs in AIM-V medium. Count the cells to determine viability and cell number.-Adjust the AIM-V volume to obtain 10 × 10^6^ cells/mL (adjust concentration according to cell number and frequency of naive T cells specific for any given antigen).-Take 10 × 10^6^ cells/mL in AIM-V and add FLT3L at final concentration of 50 ng/mL.-Distribute 250 μL/well (2.5 × 10^6^ cells) into a 48-well plate.-Incubate at 37 °C/5% CO_2_ overnight.•Day 1: Adding antigens and stimuli -Prepare GIL and ELA peptide antigens in AIM-V medium and add desired stimulating agents.-Distribute 250 μL/well of antigen and/or stimuli media to the 250 μL already present in the wells from Day 0 by gently rimming across the top of the well not to disturb the adherent cells.-Incubate at 37 °C/5% CO_2_.

Notes:○Use AIM-V medium on Day 0 and Day 1. Use R10 for medium replacement on Day 4 and Day 7.○Antigens and stimuli are prepared at twice the desired final concentration as their concentration is diluted by half upon addition to the cell suspension. •Day 2: Add 10% of FCS to each well.•Day 4 and Day 7: Remove 250 μL of old medium and add 250 μL of fresh R10.•Day 10–11: Collect cell suspension and wash with PBS to prepare for the staining with antigen-specific tetramers and antibodies.

## 2. Scope of the Method in Diagnostic Settings

### 2.1. Monitoring Antigen-Specific T Cell Responses After Vaccination or Viral Infections

The use of the acDC assay by Mallone and colleagues to monitor vaccine-induced T cell responses has been validated in several vaccine contexts. Using this assay, the authors were able to detect modified vaccinia Ankara-specific IFN-γ-producing T cells and flu-specific CD4^+^ and CD8^+^ T cells in vaccinated individuals, with higher sensitivity compared to conventional methods [21]. In a cohort of melanoma patients vaccinated with autologous DC-derived exosomes loaded with a DP4-restricted MAGE247-258 peptide, this assay enabled the identification of MAGE-specific responses in 2 out of 10 patients, responses that were undetectable using assays such as ELISpot or tetramer staining [21]. More recently, assessment of vaccine-specific memory T cells was performed in middle-aged and older COVID-19 vaccinees using extended culture with SARS-CoV-2 spike peptides in the presence of IL-2 and IL-7 over 24 days [30]. Older individuals exhibited limited spike-specific CD4^+^ and CD8^+^ T cells responses, although this depended on the vaccine platform, i.e., ChAdOx1-S (mRNA lipid nanoparticle vector) or BNT162b2 (adenoviral vector) vaccines.

In addition to monitoring vaccine induced T cell responses, this approach can simply be used for the tracking of CD4^+^ and CD8^+^ T lymphocyte responses in viral infections. These responses can exhibit considerable heterogeneity among individuals, influenced by various factors such as age [31,32], chronicity of infection [33,34,35,36], co-morbidities [37,38], and genetic background [39,40]. In the context of HIV-1 infection for instance, this approach enabled us to demonstrate that long term antiretroviral therapy (>10 years) had a positive effect on the functionality of HIV-1-specific CD8^+^ T cells, which displayed robust proliferative capacity and upregulation of cytotoxic factors, like granzyme B and perforin, to similar levels as those from natural controllers [41]. Collectively, these studies demonstrate the benefits of the in vitro approach to monitoring antigen-specific T cell responses following vaccination or during viral infections.

### 2.2. Assessing T Cell Priming Capacity in Cohorts

The main strength of this approach resides actually in the possibility of assessing the priming of antigen-specific naive T cells. Priming represents the initial and critical step in the initiation of T cell responses, during which a naive T cell encounters its cognate epitope presented by antigen-presenting cells (APCs), usually dendritic cells. Throughout this phase, the various signals generated by the interactions between naive T cells and APCs influence the differentiation and functionality of the T cells. These signals encompass the crucial interaction of the peptide–MHC complex with the TCR and the engagement of co-stimulatory or co-inhibitory receptors, as well as the effects of secreted cytokines [42,43]. This process may be altered in contexts of compromised immunity, such as old age or chronic virus infections. The study of T cell priming capacity is thus important to obtain insights into impairments of antigen-specific CD8^+^ T cell responsiveness in these contexts. Our approach enabled us to assess the influence of advanced age on T cell priming capacity. Compared to middle-aged adults, older individuals (>70 years) exhibited markedly reduced de novo CD8^+^ T cell responses, characterized by fewer antigen-primed T cells, which were less differentiated and showed reduced expression of effector molecules like granzyme B and perforin [44]. It is noteworthy that the frequency of in vitro primed CD8^+^ T cells correlated with the numbers of vaccine specific T cells measured ex vivo upon primary vaccination in the same donors [44].

The same strategy was applied to assess T cell priming capacity in people infected with persistent viruses like HCV or HIV-1. In HCV-infected individuals, naive CD8^+^ T cells present a decreased cell surface expression of CD5, associated with increased reactivity to TCR signaling. This hyperreactivity results in an enhanced expansion of ELA-specific CD8^+^ T cells following in vitro priming from HCV-infected individuals compared to healthy donors [45]. In contrast, people living with HIV-1 (PLWH) on ART exhibited reduced frequencies of ELA-primed CD8^+^ T cells, associated with lower counts of naive CD8^+^ T cells, compared to uninfected donors. It is worth noting that despite this numerical difference, antigen-primed CD8^+^ T cells from PLWH presented normal functional and phenotypic characteristics, in line with a qualitatively preserved status of naive T cells [46]. Altogether, these studies emphasize that this straightforward and innovative in vitro approach represents a valuable tool to explore de novo T cell responsiveness in different contexts, such as old age or chronic viral infections, and can provide valuable insights into factors impacting these responses.

The high frequency of ELA-specific naive T cell precursors in HLA-A2^+^ individuals and their stimulation with the ELA peptide facilitate the analysis of T cell priming using relatively limited numbers of PBMCs as starting materials (e.g., 2.5 million PBMCs). However, studying the priming of T cells specific to other epitopes is more challenging and less broadly applicable, as the precursor frequency for epitopes other than ELA is estimated to be one to two orders of magnitude lower [47]. It is nonetheless possible to apply the same in vitro T cell priming approach to other specificities, although this will require culture conditions with a much higher number of starting PBMCs (e.g., >50 million PBMCs), simply in order to increase the frequency of naive T cells specific for these peptides to be primed in the assays. For instance, the priming of HIV-1-specific CD8^+^ T cells from HLA-A24:02^+^ HIV-1 seronegative individual PBMCs was demonstrated using the HLA-A24:02-restricted HIV-1 epitope Nef RF10 [48]. Similarly, investigators employed in vitro assays to study priming of SARS-CoV-2-specific T cells from HLA-A2^+^ SARS-CoV-2 unexposed individuals of various age groups using virus-derived peptides. It was observed that older individuals exhibited impaired SARS-CoV-2-specific T cell priming capacity, characterized by a diminished response when compared to younger donors [49].

### 2.3. Testing the Effects of Immunomodulators on T Cell Priming

Adjuvants are essential components of vaccine design and formulation. By modulating cytokine and chemokine production, inflammasome activation, autophagy, and other immune-protective mechanisms, they enhance the magnitude, quality, and durability of vaccine responses [50]. Adjuvants are broadly classified into two main groups: delivery systems and immune potentiators [51]. Delivery systems primarily serve as antigen carriers, promoting local inflammation and recruiting immune cells to the injection site. In contrast, immune potentiators directly activate immune cells, such as antigen-presenting cells, creating an inflammatory environment essential for an optimal adaptive immune response. Gaining a deeper understanding of adjuvant mechanisms is crucial for optimizing vaccine efficacy. Our simple and straightforward in vitro approach enables us to study the adjuvanticity of various immune potentiators. We could examine the effect of several pattern recognition receptor (PRR) ligands, including Toll-like receptors (TLRs), nucleotide oligomerization domain (NOD)-like receptors (NLRs), and STING, as well as various cytokines or metabolic factors on antigen-specific CD8^+^ T cell priming or boosting.

For instance, we demonstrated that ssRNA40, a TLR8 ligand, was particularly effective in promoting CD8^+^ T cell expansion and acquisition of strong effector functions [29]. ELA-specific CD8^+^ T cells primed in the presence of TLR8L displayed potent killing capacity towards melanoma cell lines along with high levels of polyfunctionality (expressing TNF, MIP-1β, IL-2 and IFN-γ) compared to those primed with TLR4L. Similarly, PamadiFectin, a dual TLR2 and TLR7 ligand that promotes DC maturation and balances Th1/Th2 responses, enabled the induction of robust cellular responses in vitro [52]. We could also show that a chimeric TLR7L/NOD2L molecule, which strongly activates TLR7 and induces a pronounced proinflammatory response, effectively enhanced priming of antigen-specific CD8^+^ T cells [53]. Together with the promising characteristics of NOD2L as a potential mucosal adjuvant [54], these findings support the potential application of TLR7L/NOD2L in the development and design of mucosal vaccines. In contrast, CpG ODN 2006, a potent TLR9 activator, failed to induce an effective cellular response using our approach, despite it was demonstrated to have adjuvanticity in animal studies [55]. Beyond TLR ligands, we also investigated the adjuvant properties of cGAMP, a natural STING ligand. cGAMP appears even more effective than TLR ligands in priming highly functional antigen-specific CD8^+^ T cells in vitro, and its potency was further confirmed in in vivo mice models [56,57,58]. Given the growing interest in STING pathway adjuvants, this in vitro approach could be employed to investigate and compare the effects of novel STING agonists, like amidobenzimidazole (ABZI)-based compounds [59], non-nucleotide STING agonist MSA-2 [60], or small-molecule agonist C53 [61] on the induction of T cell responses. Overall, these studies strongly support the application of PRR ligands for T cell-based vaccines and immunotherapies.

Cytokines are potent immunostimulatory molecules, which have essential role for optimal T lymphocyte activation [62]. In line with several in vitro or in vivo studies, our system confirmed that IL-12 or type I interferons significantly enhanced the quality of cellular responses, as evidenced by the priming of ELA-specific CD8^+^ T cells with improved lytic functions [29,56]. In contrast, although IL-18 is known to promote effector functions in CD8^+^ T cells in murine models, it inhibited both the de novo induction and memory recall of antigen-specific T cells in our in vitro human system [63]. Notably, elevated serum levels of IL-18 correlated with impaired T cell-mediated responses in severe COVID-19 patients, thus suggesting that excessive accumulation of proinflammatory cytokines may hinder the activation of cellular responses.

Targeting cell metabolism to enhance immunotherapy has recently gained significant attention in vaccinology [64,65]. A growing body of evidence has highlighted the link between cellular metabolic programs and the T cell lifecycle, from naive precursors to effector cells [66]. Therefore, a deeper understanding of the metabolic mechanisms shaping T cell development could facilitate the design of more effective strategies to enhance vaccine efficacy. The use of specific metabolic inhibitors in our system resulted in reduced ELA-specific CD8^+^ T cell priming, suggesting that both mTOR-mediated glycolysis and autophagy played important roles in supporting T cell proliferation and functionality upon antigen mediated priming [67]. In the context of aging, reducing lipid component levels with L-carnitine could restore partially the cytolytic functions of cells primed from older subject PBMCs. Further in vitro experiments confirmed that inducing fatty acid oxidation using Rosiglitazone, a selective PPARγ agonist, could rescue the impaired priming capacity of old naive CD8^+^ T cells by reducing excessive neutral lipid levels and promoting anti-apoptotic effects [68]. These data suggest that lipid-modifying drugs may serve as potential immunomodulators to enhance immune response in immunotherapy and vaccines.

### 2.4. Testing Viruses or Vaccines on the Induction of Antigen-Specific T Cells

This in vitro approach can also be used to assess and compare stimulatory properties of more complex stimuli than peptides, such as whole viruses or vaccine platforms. For instance, innate immune sensing differs between HIV-1 and HIV-2 [69,70], which may result in distinct stimulation of CD8^+^ T cells. However, it is difficult to compare directly the impact of these two viruses on the induction of CD8^+^ T cell responses since they share only a 50% nucleotide sequence homology and no single epitope is known to be conserved across HIV-1 and HIV-2 [71,72]. To overcome this limitation, the Melan-A/Mart-1 sequence (i.e., ELA epitope) was inserted into HIV-1 and HIV-2 laboratory strains, which were then used to prime in vitro ELA-specific CD8^+^ T cells from total PBMCs of HLA-A2^+^ uninfected donors. This approach enabled us to compare the influence of HIV-1 or HIV-2 on T cell priming in an unbiased manner. Upon priming, we observed a stronger expansion of ELA-specific CD8^+^ T cells in the presence of HIV-2 compared to HIV-1. HIV-2-primed CD8^+^ T cells were also characterized by higher expression of the transcription factor T-bet and the cytotoxic factors granzyme B and Perforin. This superior capacity of HIV-2 to prime naive CD8^+^ T cells could be related to the production of type I IFNs, which was increased owing to the capacity of HIV-2 to infect dendritic cells [73].

More recently, a similar in vitro approach was applied to the identification and selection of HCV immunogenic epitopes for the development of HCV vaccines. In vitro priming of naive CD8^+^ T cells was achieved by stimulating total PBMCs from healthy HLA-A2^+^ donors with a DC maturation cocktail and irradiated hepatic cells lines expressing the HCV NS3 protein. The induction of HCV-specific CD8^+^ T cells was revealed after 10 days of culture, when T cells were restimulated with long peptides covering the full-length NS3 protein. This procedure enabled the identification of three HCV peptides, potential candidate epitopes for the design of HCV vaccines [74]. This approach may also be exploited for the discovery and validation of tumor-associated antigens [75,76]. On the long road to T cell vaccine development, simple in vitro assays present also the advantage to help screening potential vaccine platforms in a straightforward manner using human cells, before proceeding to in vivo models and clinical trials [77]. For instance, we are currently investigating the capacity of lipid nanoparticle (LNP) vaccines loaded with mRNA encoding the ELA epitope or viral epitopes and proteins from influenza, SARS-CoV-2, or HIV-1 to prime or boost antigen-specific CD8^+^ T cell responses. This in vitro model can thus provide a rapid overview of promising candidates that are suitable for vaccine development.

## 3. Limitations and Perspectives

In conclusion, the in vitro approach described here offers a valuable platform for monitoring the induction of human cellular immune responses in various vaccine contexts. It complements conventional assays such as ELISpot, ICS, and peptide–MHC multimer complexes, which provide ex vivo and more quantitative assessments of T cell responses, and can even be used in combination with them. Its usefulness lies in the T cell expansion step, which enables detection of low frequency T cells and allows testing of immunomodulatory molecules or vaccine candidates in diverse human settings. The simplicity and straightforward design of the system present both strengths and limitations. Since it relies on PBMC samples from a range of human donors, with varying frequencies of antigen-specific precursor cells, there is inherent variability between donors and even between assays. This variability is especially pronounced when priming T cells from the naive compartment. To obtain meaningful and statistically significant results when comparing groups or evaluating molecular effects, a minimum of 10–20 independent assays is generally required.

Despite its in vitro nature, this approach has revealed compelling parallels with in vivo T cell induction. For instance, the frequency of CD8^+^ T cells primed in vitro using the ELA peptide as a model antigen to assess priming efficacy in older individuals was found to correlate with de novo T cell responses observed following primary vaccination in the same subjects [44]. Moreover, the gene expression profile of ELA-specific CD8^+^ T cells primed in vitro in the presence of type I interferon inducers closely resembled that of HIV-specific CD8^+^ T cells isolated ex vivo from natural HIV controllers [73]. Additionally, the reduced in vitro priming of spike-specific CD8^+^ T cells from PBMCs of older individuals echoed real-world observations following SARS-CoV-2 infection and vaccination [49,63]. Although direct comparative studies assessing in vitro versus in vivo T cell responses targeting the same epitopes in the same individuals are still lacking, growing evidence supports the relevance of this in vitro model for generating biologically meaningful insights applicable to in vivo contexts. However, it still does not fully capture the complexity of T cell priming or boosting processes that occur at the injection site or within the lymph nodes [78]. Organoid models, which more closely mimic in vivo conditions, may provide deeper insights into cellular responses, particularly if they can be integrated with antigen-specific T cell induction possibilities [79]. Interest in the development of 3D models is increasing, with several studies highlighting their potential for investigating T cell biology and responses [80,81,82,83]. Lymphoid organ-on-chip models are also being successfully developed; however, these have predominantly focused on humoral immune responses thus far [84]. A key challenge for such systems remains their limited ability to support optimal culture conditions over the extended periods (typically 10–15 days) required for dynamic evaluation of T cell priming.

It is also important to emphasize that our model primarily focuses on the early phase of T cell differentiation due to the short-term cell culture. While we can assess the boosting of T cell responses driven by pre-existing memory cells—for instance, those specific to viral antigens—we are unable to track the long-term evolution of in vitro primed or boosted T cells into memory cells. Furthermore, with regard to T cell priming, this approach is generally limited to CD8^+^ T cells and the use of the ELA epitope as model antigen. Expanding this system to include immunodominant MHC-II-restricted epitopes capable of priming naive CD4^+^ T cells would significantly broaden its applications. Despite these limitations, this approach helps providing a comprehensive characterization of antigen-specific CD8^+^ T cell responses using human materials, thereby adding to existing assays and opening new possibilities for vaccine development and immunomonitoring. It enables the testing of various molecule or adjuvant combinations for T cell induction, which is crucial for addressing immune dysfunctions that arise in specific settings like aging. This paves the way for developing tailored vaccines to better protect vulnerable populations, including older individuals.

## Figures and Tables

**Figure 1 vaccines-13-00729-f001:**
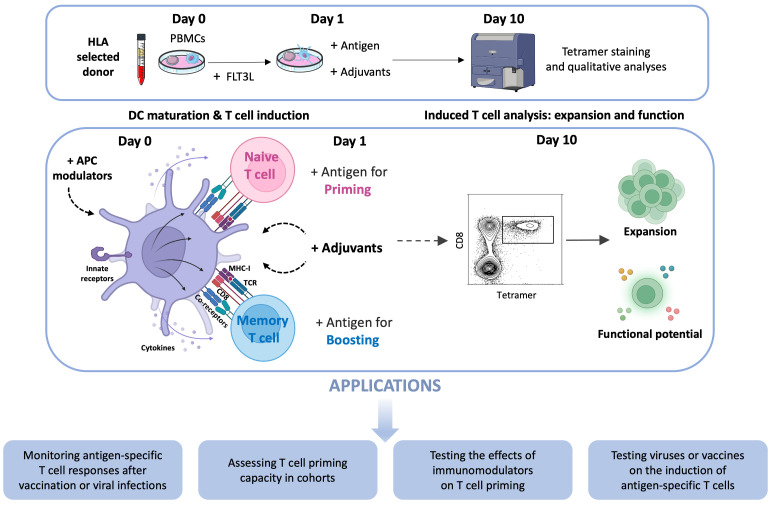
Graphical representation of the protocol for the in vitro induction of antigen-specific T cells from unfractionated PBMCs and its potential applications. The individual steps are illustrated schematically; detailed experimental conditions are described in the section below. The violet cell represents an antigen-presenting cell (APC). The flow cytometry dot plot shows tetramer^+^ cells within the CD8^+^ T cell population. Green cells indicate the induced antigen-specific T cells and the colored dots represent their functional potential.

## Data Availability

No new data were created or analyzed in this study. Data sharing is not applicable to this article.

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
