# Peer review of "An In Vitro Approach to Prime or Boost Human Antigen-Specific CD8^+^ T Cell Responses: Applications to Vaccine Studies"

_vaccines, 2025, doi:10.3390/vaccines13070729_

Round 1
Reviewer 1 Report
Comments and Suggestions for Authors
This manuscript presents a comprehensive review and methodological overview of an innovative in vitro system for priming and expanding human CD8+ T cell responses using PBMCs and dendritic cell stimulation. The approach is clearly described and substantiated with several applications, ranging from vaccine immunomonitoring to adjuvant screening. The review is timely and relevant, particularly given the increasing interest in T cell-based vaccines and the limitations of current animal models and ex vivo techniques. Overall, the manuscript is well written, informative, and provides valuable insight into translational immunology and vaccinology.
Major concerns:
- Authors have mentioned several applications qualitatively, it would enhance the manuscript to include more quantitative comparisons (e.g., frequency or fold-expansion of CD8+ T cells) between different conventional techniques such as ELISPOT or ICS.
- Also, a more in-depth discussion of how this in vitro model correlates with in vivo outcomes (e.g., predictive validity for vaccine efficacy) would be valuable.
- It would be great if authors throw some light on standardization assay and reproducibility i.e. how it is managed across laboratories would strengthen translational relevance.
- Also Author should include small section on organoid models and STING pathway adjuvants and related recent references
Minor Concerns:
- Figure 1 legend should be in detail explaining everything.
- Line 103-104: Authors wrote this method presents several advantages. It would be great if they could explain some of them briefly.
- Line 153, 158, 168 - Authors mentioned 250ml in 48 wells. I think it's typographical mistake - It should be 250µL (Check throughout the manuscript)
- Line 168 - Authors wrote "Day 4 & Day 7: Change medium by replacing half of the medium with 250ml of R10" can you please explain this? what do you mean by replacing half medium? How do you do that?
The manuscript is suitable for publication following these revisions to strengthen clarity, expand on quantitative data, and further contextualize the assay’s translational implications.
Reviewer 2 Report
Comments and Suggestions for Authors
The manuscript « An in vitro approach to prime or boost human antigen-specific 2 CD8+ T cell responses Applications to vaccine studies » by Nguyen et al., is a theoretical manuscript addressed to discuss the genesis of a procedure for quantitative and qualitative readouts of primed antigen-specific CD8+ T cells. This manuscript presents updated literature and discusses some methodological aspects that have been tested in various models as well as in in vitro assays. The manuscript is divided into several chapters, the first one is dedicated to establishing the need for human-based approaches and assays in vaccinology. It is established that in current vaccination systems, efficiency is mainly directed to the humoral regulation of the response through the activation of CD4+ cells and, to a lesser extent, CD8+ cells, although CD8+ T cells are nonetheless essential for efficient viral clearance, through the induction of apoptosis through the recognition of peptides presented in the context of MHC1. General aspects of a multipurpose CD8+ T cell induction approach are proposed and experimentally suggested to make this process of activation of antigen-specific CD8+ cells efficient through recruitment or selection of dendritic cells.
A general list (and outline) of the description of the standard protocol is presented, intended as a summary of standard procedures for the isolation and activation of cell groups. A section is proposed summarizing various processes for monitoring antigen-specific T cell responses after vaccination or viral infections, including the effects of immunomodulators on T cell priming.
The review of these processes is interesting but does not provide novel processes or new strategies. It would be interesting to present some results or experimental evidence that would allow us to identify what truly offers a significant advantage for monitoring the induction of cellular immune responses across different vaccine contexts.
Reviewer 3 Report
Comments and Suggestions for Authors
The authors introduced an innovative in vitro method for priming and monitoring human antigen-specific CD8+ T cell responses from unfractionated PBMCs, using the Melan-A/MART-1 epitope (ELA) as a model antigen. The approach enables versatile applications, including vaccine immunomonitoring, assessing T cell priming capacity in aging or chronic infections, and screening adjuvants (e.g., TLR8/STING ligands) and vaccine platforms (e.g., mRNA-LNPs). This standardized system offered a valuable tool for preclinical vaccine development and immune response analysis.
Here are some questions and suggestions:
- The authors should further discuss the limitations of this new method, particularly in comparison to conventional CD8 CTL assessment techniques, eg., ELISpot, ICS, peptide-MHC multimers.
- The reliance on the high-frequency Melan-A/MART-1 (ELA) epitope may limit generalizability. The paper should address whether the method works as effectively for lower-frequency epitopes.
Round 2
Reviewer 2 Report
Comments and Suggestions for Authors
The manuscript, as indicated in the first review, in this new versión the manuscript is better improved, contains valuable information that allows us to identify various alternatives for understanding the bases of the response to various immunogens and the various approaches to studying it. However, when the authors try to justify their protocol in the related chapter "Genesis of a multipurpose CD8+ T cell induction approach," in the first part of the justification and genesis, the need to establish protocols that allow specific identification of specific activation is clear. However, the description of the protocol, including the scheme without showing specific results, is somewhat difficult to understand.